# A multispecies corridor in a fragmented landscape: Evaluating effectiveness and identifying high-priority target areas

Karen E. DeMatteo[1,2]*, Orlando M. Escalante[3,4], Daiana M. Ibañez Alegre[3], Miguel A. Rinas[5], Delfina Sotorres[3,4], Carina F. Argüelles[3,4]

1 Department of Biology & Environmental Studies, Washington University in St. Louis, St. Louis, Missouri, United States of America, 2 WildCare Institute at the Saint Louis Zoo, St. Louis, Missouri, United States of America, 3 Grupo de Investigación en Genética Aplicada (GIGA), IBS–Nodo Posadas, Universidad Nacional de Misiones (UNaM)–CONICET, Posadas, Misiones, Argentina, 4 Facultad de Ciencias Exactas, Departamento de Genética, Químicas y Naturales, UNaM, Posadas, Misiones, Argentina, 5 Ministerio de Ecología y Recursos Naturales Renovables, Posadas, Misiones, Argentina

☯ These authors contributed equally to this work.
* karendematteo@outlook.com

**Data Availability Statement:** Georeferenced data cannot be shared publicly due to provincial law in

## Abstract

While Misiones, Argentina contains one of the largest remnants of Upper Paraná Atlantic Forest ecoregion, one of the world's biodiversity hotspots, only ~50% of this native forest is protected. Each protected area is at risk of becoming an island of native forest surrounded by a matrix of altered habitats due to ongoing land conversion. In an effort to maximize long-term connectivity between existing protected areas, DeMatteo [1] used a multifaceted cost analysis to determine the optimal location for the region's first multispecies corridor using noninvasive data on jaguars (*Panthera onca*), pumas (*Puma concolor*), ocelots (*Leopardus pardalis*), southern tiger cats (*Leopardus guttulus*), and bush dogs (*Speothos venaticus*). This work builds on this framework by integrating new field data that broadens the scope of species-specific data across the region's heterogeneous landscape, which varies in vegetation, disturbance, human proximity, and protective status. In addition, two different land use layers are compared across the distributions of the five carnivores, the overlap in their independent distributions, and their relationship to the multispecies corridor. Interpretation of these land use data to species-specific habitat suitability goes beyond DeMatteo [1], with a subdivision of suitability into marginal and optimal areas. This refined scale allows a reanalysis of key areas in the multispecies corridor, where connectivity was previously defined as at highly-at-risk, allowing for a more directed development of management strategies. These analyses and their interpretation extend beyond northern-central Misiones, as the threats are not unique to this region. The need to develop management strategies that balance human-wildlife needs will continue to grow as humans expand their footprint. The techniques applied in this analysis provide a way to identify key areas that require specific management strategies, either through restoration, protection, or a combination of both.

the province of Misiones, Argentina. The species locations are protected due to threats of poaching and their endangered status. However, requests for the data can be made through Cristina Buhler (dirbio_proy_inv@misiones.gov.ar) at the Ministerio de Ecologia in Dirección de Biodiversidad.

**Funding:** This study was supported by Chester Zoo, the Conservation, Food and Health Foundation, Eppley Foundation for Research, Fresno Chaffee Zoo Wildlife Conservation Fund, Jaguar Conservation Fund (Woodland Park Zoo), Kickstarter, Little Rock Zoo Foundation, National Geographic Society, Palm Beach Zoo Conservation & Science Program, Paris Zoo, Phoenix Zoo Conservation & Science Program, Zoo Atlanta (Georgia AAZK, Reeder Conservation & Science Program, & Quarters for Conservation), Riverbanks Zoo and Garden, Sequoia Park Zoo, and the New England Conservation Committee, all awarded to KEM.

**Competing interests:** The authors have declared that no competing interests exist.

# Introduction

As the human footprint continues its expansion worldwide [2,3], there is a need to develop management strategies that balance human-wildlife needs in the face of habitat fragmentation and varied human activities across the landscape [4–6]. This includes understanding the effect of expanding human demands on the landscape relative to the long-term survival of biodiversity, habitat integrity, and ecosystem services [6–10]. One approach is the use of a Geographic Information System (GIS) to conduct spatial analyses that identify wildlife corridors or areas that ensure connectivity across fragmented landscapes, while minimizing the potential for human-wildlife conflict [11,12].

Wildlife corridors are often designed using existing protected areas as a series of "stepping stones" across the heterogeneous landscape [1,13–17], which requires understanding whether these intermediary areas between protected areas are a barrier to species movement (high ecological cost) or seen by the species as suitable, even if of lower preference (e.g., lower prey abundance, less protected cover) [18–20]. Defining this suitability requires generating a surface of resistance to animal movement, which depends on deciding which factors should be included and the specific weights that should be assigned to reflect their effect (positive or negative) on animal movement [21,22]. The method applied to develop the surface will vary depending on whether actual movement data is available (e.g., resource selection function) [23–26] or not (e.g., species occurrence, density estimates, expert opinions) [27–29]. The technique to model connectivity (e.g., circuit theory, network flow, path selection, least-cost past, least-cost corridor) will vary depending on study scale and focal species [30–35]. This extends to defining the optimal or minimal width for the corridor, with the need to balance its length and width [21,26,36].

Some corridors are designed with a focus on a single species, to protect coexisting species through an umbrella effect, but this may fail to completely capture the varied ecological requirements of coexisting species and ecological processes across the landscape [32,37,38]. Others are based on multiple species, to capture the breadth of ecological requirements, but this may generate results that underestimate connectivity for species highly restricted in their movements [38–40]. A more comprehensive approach balances the trade-offs of these two methods and accounts for variation across multiple species, with information on the most restricted species used to weight final decisions [32,37]. DeMatteo [1] used this balanced method to define a multispecies corridor in northern-central Misiones, Argentina that would almost double the protected area (+400,000 ha) and secure an additional ~300,000 ha of native forest. Specifically, DeMatteo [1] defined the corridor using the breadth of species-specific habitat preferences for five carnivores [jaguar (*Panthera onca*), puma (*Puma concolor*), ocelot (*Leopardus pardalis*), southern tiger cat (*Leopardus guttulus*), and bush dog (*Speothos venaticus*)], while ensuring the most restricted species (jaguar) was effectively captured [18,41,42].

The native forest that the DeMatteo [1] multispecies corridor would protect is part of one of the largest remaining tracts of the Upper Paraná Atlantic Forest ecoregion, one of the world's biodiversity hotspots [43,44], with Misiones containing almost 1.4 million ha [45]. Despite the small size of Misiones, which occupies only 1% of the total area in Argentina, it is considered its National Capital of Biodiversity (Ley Nacional 27494), a designation that recognizes it contains >40% of the country's biodiversity. However, only ~35%, or ~500,000 ha, of this unique forest is protected in a series of areas that vary in size, adjacency, and degree of protection. The connectivity between protected areas and long-term survival of its biodiversity is threatened by many factors, including ongoing habitat conversion, forest degradation, an expanding network of roads, population growth in rural areas, and poaching [1,46–50]. The native forest outside of protected areas exists in a matrix of plantations, agriculture, and

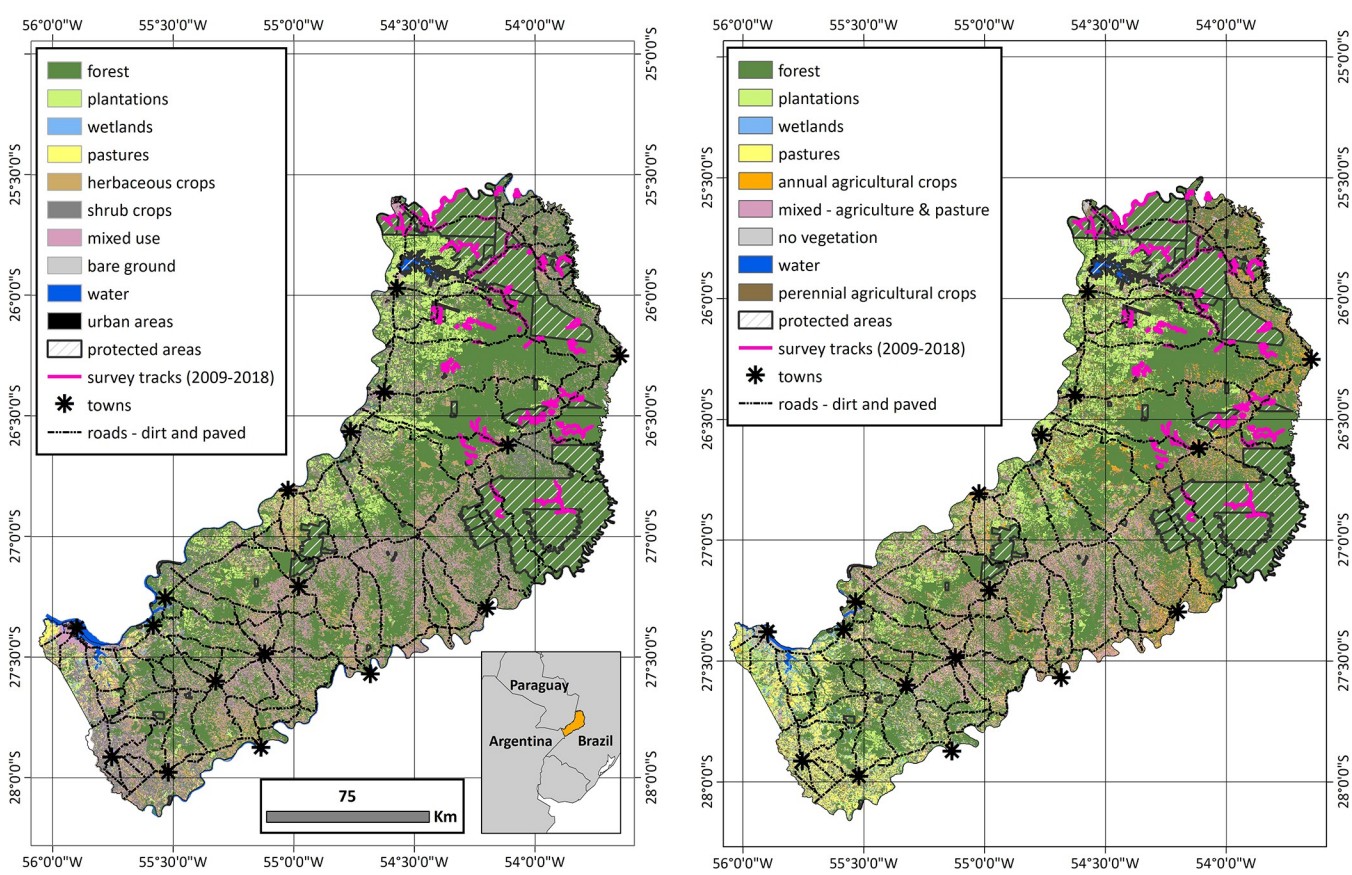

**Fig 1. Map of northern-central Misiones, Argentina with detection dog survey routes (2009 thru 2018) shown relative to protected areas, major roads, towns, and the two land-use raster grids used in spatial analyses.** a) land use was derived from a mosaic of Landsat-8 TM satellite images from 2015 [45] and b) land use was derived using the automated classification of satellite images from 2018 [53].

pastures, which continue to expand annually [1,20,45,51,52] (Fig 1). In 1999, the province of Misiones proactively developed a plan to create the Green Corridor [Corredor Verde; Ley Provincial XVI N˚60 (previously Ley Provincial N˚ 3631/99)], to protect this unique ecoregion; however, over the last two decades, on-the-ground protection of the forest, waterways, and biodiversity has been varied and irregular, with the Green Corridor primarily existing on paper.

The multispecies corridor modelled by DeMatteo [1] effectively provides a strategic starting point to make the Green Corridor a reality. However, only ~40% of the 761 carnivore scats used to model this corridor were located outside of protected areas or in those areas that are essential for connectivity between existing "stepping stones" or protected areas. While this initial data represented a first for the region in its sample collection across multiple species independent of habitat integrity, proximity to humans, and degree of habitat protection, subsequent detection dog surveys in the northern-central zone allowed the collection of an additional 324 scats from the five targeted carnivores, with almost 80% of samples located outside of protected areas. This work will build on the framework established by DeMatteo [1] integrating these new field data [26] for a comprehensive set of 1,085 scats located across the corridor's heterogeneous landscape (e.g., degraded forest, pastures, near human populations, large-scale agriculture, monoculture plantations) that can be used to ground-truth the DeMatteo [1] multispecies corridor. This expanded analysis will not only allow confirmation that the

location of the corridor is optimal but also provides insight into the effect of sample distribution (e.g., in/out of a protected area) on the efficacy of corridor modelling. These analyses will expand beyond DeMatteo [1], which looked only at suitable versus unsuitable habitat for the five carnivores, and include optimal, marginal, and unsuitable habitat quantities. These finer divisions across the corridor will be used to identify key regions in need of management strategies that promote human-wildlife coexistence [49,54,55]. This last step is essential given that ~60% of the multispecies corridor is privately owned, so balancing the need of humans to "live off" the land with the need for biodiversity to live in the heterogeneous landscape is a must for the long-term conservation of the Green Corridor [4,56–59].

## Materials and methods

### Field surveys

Data were collected during five surveys (2009, 2011, 2013, 2016, & 2018) conducted primarily during the cool season (May-August) of Misiones, Argentina. The Ministerio de Ecología y Recursos Naturales Renovables of Misiones (MEyRNR) issued all general permits related to our project in the province, collection of samples in the multiple provincial parks, and export permits of samples for genetic analyses. The Administración de Parques Nacionales of Argentina issued permits related to the collection of samples in the national park. Per the regulations of the Saint Louis Zoo's Institutional Animal Care and Use Committee, no written or verbal approval was required for this study.

Survey tracks consisted of two-lane paved roads, 1–2 lane dirt roads, established machete-cut trails, illegal hunting trails, and animal trails. The team surveyed a total of 257 unique routes and walked a total of 1,613.7 km [mean (SD) = 6.28 (3.62) km; range = 0.23–21.06 km per route; Fig 1]. The total coverage in the northern zone (992.4 km, 61.5%) was slightly higher (371.1 km, 23%) than that of the central zone (621.3 km, 38.5%). In addition to protected areas of native forest, surveys covered unique habitats including private native forests, small-scale agriculture, monoculture plantations of pine and eucalyptus, small communities with subsistence agriculture, pastures, and human-occupied areas. Of the total distance walked, nearly half (747.1 km, 46.3%) was located outside of protected areas.

All carnivore surveys were completed using the same detection dog-handler team [1,20,51,60]. Detection dogs eliminate dependence on visitation rate to a specific location (e.g., camera trap) and instead switches the focus to locating evidence (e.g., olfactory) associated with the species' natural behavior and movement patterns. Even in the rugged terrain of Misiones, Argentina, detection dogs have been able to effectively search large geographic areas and locate samples from multiple species, while ignoring samples from nontarget species that may be similar in their appearance or composition [1,20,51,60]. The training of the detection dog, swabbing of scat for genetic analyses, collection of scats, and recording of field data were carried out using the same strategies made in all surveys [1,20,51,60].

### Genetic analyses

Scat swabs were processed using DNA extraction protocols and genetic analyses detailed in previous studies [1,20,51]; however, a brief summary is provided here. DNA was extracted from two independent swabs using a Qiagen (Venlo, Netherlands) DNeasy™ DNA extraction kit following a modified protocol by Vynne [61]. To identify species, a 110-bp (171-bp with primers) carnivore-specific region of mitochondrial cytochrome *b* gene (5′-AAACTGCAGCCC CTCAGAATGATATTTGTCCTCA-3′; 5′-TATTCTTTATCTGCCTATACATRCACG-3′ [62] was amplified with a modified version of the protocols and reagents of Farrell [62] and Miotto [63]. Amplifications were performed on a MyCycler Thermal Cycler System (BioRad,

Hercules, CA) in 25-μL volumes containing 2-μL DNA extract, 1× PCR Gold buffer [Applied Biosystems, Foster City, CA] 0.3-μM forward and reverse primer, 200 μM each dNTP, 5-mM MgCl$_2$, 150-μg/mL BSA (Ambion® - Life Technologies, Grand Island, NY) and 1-U Ampli*Taq* Gold DNA polymerase (Applied Biosystems). The PCR profile consisted of 10-min denaturation at 95˚C, followed by 40 cycles at 95˚C for 30 s, 49˚C for 45 s, 72˚C for 45 s, and a final 30-min extension at 72˚C. Purified PCR products were sequenced using the ABI PRISM Big-Dye Terminator v3.1 Cycle Sequencing Kits (ABI) and analyzed in an ABI 3100 Genetic Analyzer (ABI). Sequences were edited and aligned using Lasergene Seqman 8.1 (DNASTAR, Madison, WI) and compared with reference entries in GenBank using the Basic Local Alignment Search Tool (BLAST) [64] to identify sequences from Neotropical species that had high similarity and closely-matched sample sequences.

Of the 1,451 total samples collected, species identity was confirmed in 1,085 (74.8%) (S1 Appendix). In the remaining scat swabs, species identity was not possible due to low quantity/quality DNA or urine contamination through scent marking animals. While over half of the samples were identified as southern tiger cat (n = 714, 65.8%), there were 145 (13.4%) ocelot, 76 (7.0%) jaguar, 99 (9.1%) puma, and 51 (4.7%) bush dog. While the proportion of samples found in the northern-central zone were similar in puma (48.5%: 51.5%), southern tiger cat (50.0%: 50.0%), and bush dog (56.9%: 43.%), they were more skewed in jaguar (92.1%: 7.9%) and ocelot (73.8%: 26.2%), with the elevated proportion in the northern zone likely associated with the higher levels of contiguous forest [20]. Exact sample locations are not reported or displayed per government request as a precaution to protect these threatened and endangered carnivores from targeted poaching.

## Modeling the ecological niche

MaxEnt 3.3.3.k was used to model ecological niches for the five carnivores and evaluate habitat suitability [65], as it has been reported to perform consistently better than other algorithms [66,67] and is effective at working with small numbers of presence-only samples [67]. Models were fit using auto features and default parameters for regularization multiplier (1), convergence threshold (1.0E-5), and background points (10,000) [68,69]. For each species, models were tested by randomly withholding 25% of presence localities and running 15 subsample replicates with the average output used for interpretation. We generated a logistic output, which gives the probability of species presence on a scale of 0 to 1 and has been shown to improve model performance via model calibration of output values and corresponding suitability [68].

Applying threshold or "cutoff" values to the logistic output of MaxEnt allows it to be converted to predictions that allow habitat suitability to be evaluated. First, each model was converted to a binary prediction where habitat was defined as suitable (high probability of species' presence) or unsuitable. Ecologically, values equal to or greater than the defined threshold value can be interpreted to contain cells that are predicted to be at least as suitable as those where the species was identified present. We compared the extrinsic omission rate and proportional predicted area (proxy for commission rate) at several logistic thresholds [e.g., minimum training presence (MTP), fixed cumulative value 1 (FCV1)] [65,70,71], with our criteria to have an omission rate of zero but set lower restrictions on the size of the potential predicted area. Second, the suitable habitat for each species was subdivided into marginal and optimal habitats using the maximum training sensitivity plus specificity (MTSS), a threshold noted to be most appropriate at defining optimal habitats in presence-only models [70,72,73]. Ecologically, marginal habitat corresponds to values equal to or greater than the binary threshold but less than the MTSS threshold; whereas, optimal habitat corresponds to values greater than the MTSS threshold.

## Predictor variables for the ecological niche model (ENM)

While DeMatteo [1] used a single land-use layer from 2009 [74], this work compares two data sets that have a stronger fit to the later years of data collection (2009–2018), which effectively captures the additional forest loss and land conversion across the landscape (Fig 1). Model 1 used a land use derived from a mosaic of Landsat-8 TM satellite images from 2015 [45]. Model 2 used a land use derived from the automated classification of satellite images from 2018 [53].

Using ArcMap 10.8, a neighborhood analysis with land use type used focal statistics, at a 30 m × 30 m resolution, to characterize conditions in neighborhood cells (Table 1) independently for each land use layer. The landscape heterogeneity index is based on the Shannon-Wiener diversity index of habitat type diversity, where higher heterogeneity values represent strongly anthropogenic landscapes (i.e., a great number of human modified landscapes in a small area) [75]. A neighborhood scale of 333 cells x 333 cells was used to represent a home range of ~100 km$^2$ (99.8 km$^2$) for the jaguar [76–79], puma [79,80], and bush dog [81]. A neighborhood scale of 150 cells x 150 cells was used to represent a home range of ~20 km$^2$ (20.25 km$^2$) for the ocelot [76,79,82,83] and southern tiger cat [83]. While both of these values are more conservative

**Table 1. Summary of predictor variables tested and used in the two species-specific ecological niche models generated in this work.**

| Variable | Frequency | Final model |
|---|---|---|
| **Model 1:** | | |
| Native forest | yes | yes |
| Plantations of pine & eucalyptus | yes | yes |
| Wetlands | yes | yes |
| Pastures | yes | yes |
| Herbaceous crops of tobacco & maize | yes | yes |
| Shrub crops of tea & yerba mate | yes | yes |
| Mixed crops with small patches shrubs & herbaceous plants | yes | yes |
| Naturally bare ground | yes | no |
| Water bodies (natural & artificial) | yes | no |
| Urban and infrastructure | yes | yes |
| Landscape heterogeneity index | yes | yes |
| Land use type | no | no |
| **Model 2:** | | |
| Native forest | yes | yes |
| Forest plantations | yes | yes |
| Wetlands | yes | yes |
| Pastures | yes | yes |
| Annual agricultural crops | yes | yes |
| Mix of agriculture and pastures | yes | yes |
| No vegetation | yes | yes |
| Water | yes | no |
| Perennial agricultural crops | yes | yes |
| Landscape heterogeneity index | yes | yes |
| Land use type | no | no |

Model 1 used land use derived from a mosaic of Landsat-8 TM satellite images from 2015 [45] and Model 2 used land derived using the automated classification of satellite images from 2018 [53]. The list includes all predictor variables used in the development of the species-specific ecological niche models and whether the variable was used in the final model.

than DeMatteo [1], which used home ranges of 100 km$^2$ and 150 km$^2$ for all five carnivores, they are believed to provide a more accurate measure given the scale of these analyses.

The initial species-specific ecological niche models were run with all defined predictor variables, 12 in Model 1 and 11 in Model 2 (Table 1). Evaluation of jackknife tests of variable importance and response curves for individual variables eliminated select variables from each final model, due to either no effect or a negative effect on model performance. Both models eliminated land use type and water. Model 1 also eliminated bare ground. Interestingly, Model 2 maintained bare ground in the final model; however, this classification appears more similar to the urban and infrastructure category in Model 1. This evaluation used the regularized training gain and test gain generated by MaxEnt, which accounts for dependency among predictor variables and compares the effect of a specific feature by itself with a model of all features except that single feature. Efficacy of the nine selected predictor variables was evaluated by MaxEnt jackknife tests using test gain and area under the ROC curve (AUC) on test data, with the latter providing a threshold-independent measure of overall model accuracy [84].

## Potential species richness (PSR)

The PSR, which is obtained by combining the five species-specific ENMs, identifies areas where none (value = 0) or all (value = 5) of the carnivores overlap in habitat defined as suitable [85]. PSR highlights areas where multiple species have suitable habitat and quantifies species number. This degree of overlap is an additional method to confirm range-restricted species and identify areas where habitat restoration could result in increased PSR.

## Identification of regions in need of management strategies

The five species-specific ENMs with suitable areas defined as marginal and optimal were combined to generate a PSR and identify areas in need of management strategies. To simplify the analyses and interpretation, data were grouped into six classes: 0 species or unsuitable habitat, 1–3 species with marginal habitat in need of restoration, 4–5 species with marginal habitat in need of restoration, 1–3 species with optimal habitat in need of restoration and protection, 4 species with optimal habitat in need of protection, and 5 species with optimal habitat in need of protection.

# Results

## Final ENM evaluation

Both models had AUC values that indicated a high accuracy in discriminating areas of species' presence and absence [86]. In Model 1, the AUC values were ≥ 0.87 for test data (jaguar = 0.96 ±0.01 (SD), puma = 0.92±0.02, ocelot = 0.89±0.02, southern tiger cat = 0.87±0.01, bush dog = 0.88 ±0.03). In Model 2, the AUC values were ≥ 0.90 for test data (jaguar = 0.96±0.01 (SD), puma = 0.92±0.02, ocelot = 0.92±0.02, southern tiger cat = 0.90±0.01, bush dog = 0.91±0.02).

Each model was converted to a binary prediction of suitable and unsuitable habitat (Fig 2). In Model 1, MTP best fit the defined threshold criteria in puma (0.0258), ocelot (0.0415), and southern tiger cat (0.0055), while FCV1 was determined to be the best fit in jaguar (0.0145) and bush dog (0.021). In Model 2, the results were similar, with MTP having the best fit for the defined threshold criteria for puma (0.0376), ocelot (0.027), and southern tiger cat (0.00235), FCV1 was determined to be the best fit in the jaguar (0.0234) and bush dog (0.0235). While FCV1 was determined to be the less conservative choice in jaguar and bush dog for both models, it was considered the better choice because it identified the maximum potential areas possible while still maintaining a zero-omission rate for both training and test data.

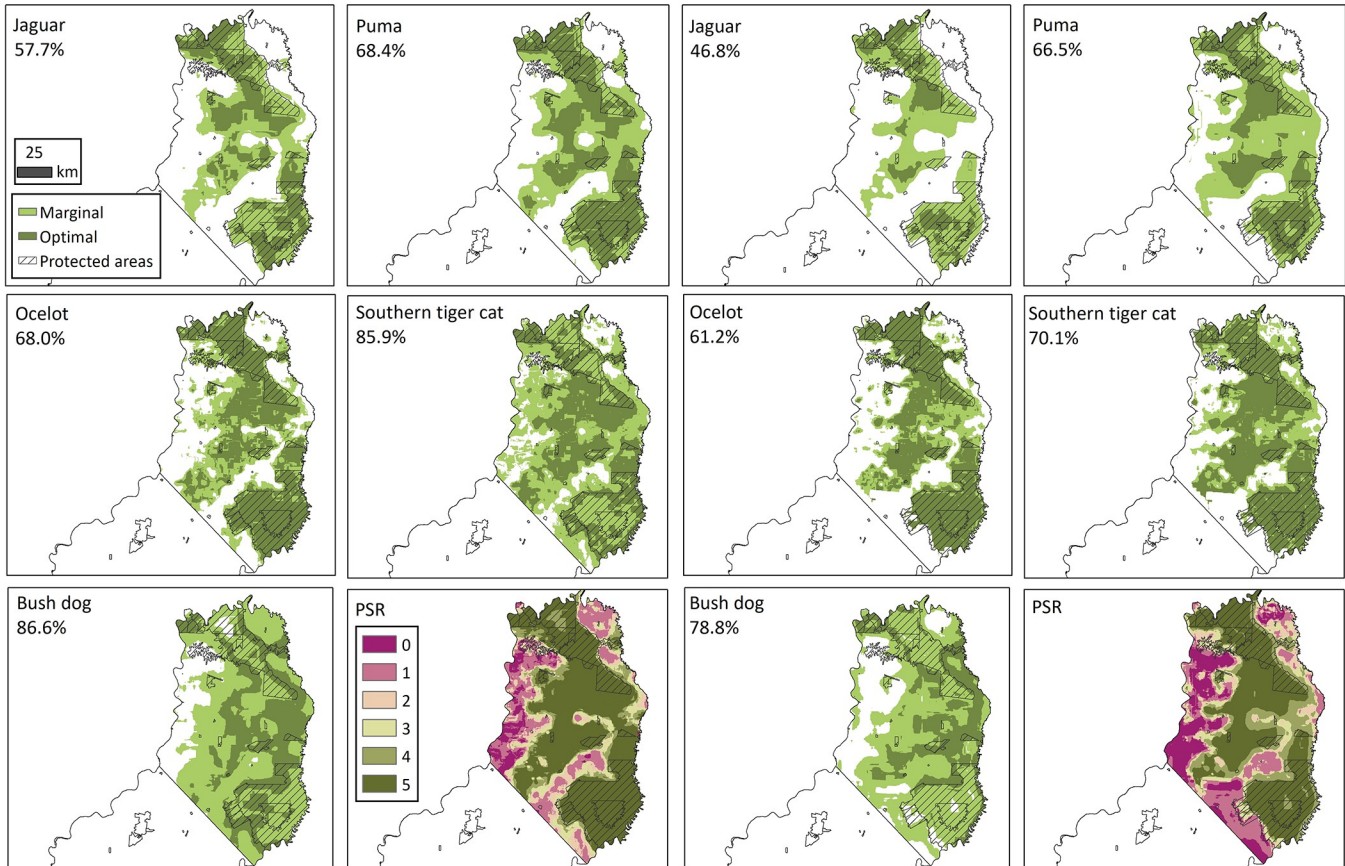

**Fig 2.** A summary of the proportion (%) of marginal and optimal habitat defined as suitable in the northern-central zone for Model 1 (left) and Model 2 (right), with these two measures equal to the binary prediction of proportion (%) of total suitable habitat. Included are the five species-specific ecological niche models and the potential species richness (PSR) that quantifies the overlap (0–5 species) among those areas defined as suitable for each carnivore. The latter is only represented as a binary prediction.

When binary threshold values were assigned and the area was restricted to the northern-central zone of Misiones (i.e., where the surveys occurred), the jaguar was identified as the most restricted species (Table 2). However, there were differences among the models with Model 1 (57.7%) having a higher total area defined as suitable habitat for the jaguar compared to either Model 2 (46.8%) or DeMatteo [1] (52.3%; Table 2). As with the jaguar, the other four carnivores had higher values in Model 1 compared to DeMatteo [1]. While the ocelot had only a minimal difference between these models (0.5%), there was a steady increase in this difference in puma (1.9%), southern tiger cat (3.6%), jaguar (5.4%), and bush dog (12.5%; Table 2). In contrast, the pattern of Model 2 having a lower proportion of total suitable habitat than Model 1 or DeMatteo [1] held true for southern tiger cat and ocelot, while puma and bush dog had equal or higher proportions of total suitable habitat (Table 2). When these values were averaged across the five carnivores, Model 1 had the highest average proportion of suitable habitat (73.3%), while Model 2 (64.7%) fell below DeMatteo [1] (68.5%).

Despite this range in the proportion of suitable habitat among the five carnivores in Model 1 (range = 57.7–86.6%) and Model 2 (range = 46.8–78.8%), all had relatively high proportions of forest in their potential distributions, with only an ~11% spread or difference (Table 3). In both models, jaguar and ocelot had the highest proportions of forest, followed by puma, southern tiger cat, and bush dog. Like DeMatteo [1], puma, southern tiger cat, and bush dog had slightly higher levels of modified habitats thanthe jaguar and ocelot (Table 3). Breaking down

**Table 2. The proportion (%) of suitable habitat, as determined with species-specific ecological niche models (ENM), across the ~1.5M ha in the northern-central (N-C) zone in Misiones, Argentina, located outside of protected areas, and captured by ~400,000 ha multispecies biological corridor modelled by DeMatteo [1].**

| | ENM | N-C zone ALL | | | N-C zone OUTSIDE PA | | | Corridor | | |
|---|---|---|---|---|---|---|---|---|---|---|
| | | Total | Marginal | Optimal | Total | Marginal | Optimal | Total | Marginal | Optimal |
| **Jaguar** | Model 1 | 57.7 | 32.5 | 25.2 | 30.2 | 18.9 | 11.3 | 33.0 | 18.9 | 14.1 |
| | Model 2 | 46.8 | 33.3 | 13.5 | 21.8 | 17.3 | 4.5 | 55.3 | 42.3 | 13.0 |
| | DeMatteo [1] | 52.3 | — | — | 25.4 | — | — | 13.9 | — | — |
| **Puma** | Model 1 | 68.4 | 31.7 | 36.7 | 39.9 | 24.51 | 15.4 | 32.6 | 17.4 | 15.2 |
| | Model 2 | 66.5 | 39.5 | 27.0 | 38.5 | 27.3 | 11.2 | 35.3 | 21.4 | 13.9 |
| | DeMatteo [1] | 66.5 | — | — | 37.6 | — | — | 21.0 | — | — |
| **Ocelot** | Model 1 | 68.0 | 25.9 | 42.1 | 38.7 | 17.2 | 21.5 | 30.5 | 11.5 | 19.0 |
| | Model 2 | 61.2 | 19.6 | 41.6 | 33.6 | 16.9 | 16.7 | 35.2 | 14.6 | 20.6 |
| | DeMatteo [1] | 67.5 | — | — | 38.3 | — | — | 21.1 | — | — |
| **Southern tiger cat** | Model 1 | 85.9 | 47.3 | 38.6 | 56.6 | 33.7 | 22.9 | 28.1 | 9.8 | 18.3 |
| | Model 2 | 70.1 | 22.4 | 47.7 | 42.4 | 19.1 | 23.3 | 33.8 | 10.4 | 23.4 |
| | DeMatteo [1] | 82.3 | — | — | 52.9 | — | — | 24.3 | — | — |
| **Bush dog** | Model 1 | 86.6 | 52.9 | 33.7 | 58.6 | 37.3 | 21.3 | 29.7 | 13.9 | 15.8 |
| | Model 2 | 78.8 | 54.1 | 24.7 | 51.6 | 33.7 | 17.9 | 30.9 | 16.9 | 14.0 |
| | DeMatteo [1] | 74.1 | — | — | 45.4 | — | — | 22.5 | — | — |

The N-C zone ALL refers to the proportion (%) of suitable habitat across the total area in the N-C zone of Misiones, including overlap with existing protected areas (~450,000 ha). In contrast, the N-C Zone OUTSIDE PA refers to the proportion (%) of total suitable habitat excluding areas that overlap with existing protected areas. The corridor refers to the proportion (%) of total suitable habitat captured by the DeMatteo [1] corridor. Each species has three levels of suitability reported: Total suitability, as defined by the binary threshold, plus this area subdivided into marginal and optimal habitat. Each species has three ENMs reported: Model 1, Model 2, and the model generated by DeMatteo [1]. With the latter, no marginal or optimal habitat are available for comparison.

these proportion of suitable habitat into the proportion located outside of protected areas allows insight into its sensitivity to ongoing changes in the landscape (Table 2). The average proportion across all five species showed little variation, with Model 1 the highest (44.8%) and Model 2 (37.6%) falling slightly below DeMatteo [1] (39.9%). Model 1 had a higher proportion of suitable area outside of protected areas in all species compared to Model 2 (Table 2). The jaguar had the lowest proportions of suitable habitat outside of protected areas (30.2% and 21.8%, respectively). The puma and ocelot were similar with slightly lower levels outside of protected areas, with Model 2 (33.6%) showing the ocelot as more restricted to protected areas. The southern tiger cat and bush dog had higher proportions of suitable habitat outside of protected areas (>50%), with the exception of Model 2 (42.4%) with the southern tiger cat.

When these binary thresholds of suitable habitat were further subdivided into marginal and optimal habitat (Fig 2), the jaguar (10% and 14.3%) and bush dog (19.2% and 29.4%) had higher proportions of marginal habitat in both models, which contrasted with the ocelot (16.2% and 22%) that had higher proportions of optimal habitat in both models (Table 2). The puma and southern tiger cat showed a varied pattern, with the proportion of marginal and optimal switching between models (Table 2). When these proportion of marginal and optimal habitat were divided into the proportion located outside of protected areas, the jaguar, puma, and bush dog had higher portions of marginal habitat located outside of protected areas, while the ocelot and southern tiger cat were varied in their proportions (Table 2).

## Potential species richness

These similarities and differences among the five carnivores in their proportion (Table 2) and type (Table 3) of suitable habitat help explain the PSR (Fig 2; Table 4). The five species-specific

**Table 3. Proportion (%) of habitat relative to the total area defined as suitable in the species-specific ecological niche models (ENM) across the ~1.5M ha in the northern-central (N-C) zone in Misiones, Argentina.**

| | Model 1 | | | | | Model 2 | | | | | | DeMatteo [1] | | | | | |
| | Native forest | Plantations | Mixed crops | Shrub crops | Herbaceous crops | Native forest | Forest plantations | Pastures | Annual agriculture crops | Mix of agriculture & pastures | Perennial agricultural crops | Forest | Tree plantations | Agriculture | Mixed use | Urban | Pasture |
|---|---|---|---|---|---|---|---|---|---|---|---|---|---|---|---|---|---|
| **Jaguar** | 89.4 | 6.3 | 1.6 | 1.3 | 1.2 | 89.4 | 6.7 | 1.3 | — | — | 1.5 | 77.2 | 9.9 | 5.6 | 2.5 | 2.1 | — |
| **Puma** | 85.8 | 6.9 | 3.1 | 2.4 | 1.7 | 82.7 | 6.9 | — | 2.2 | 3.6 | 3.2 | 74.4 | 9.6 | 8.5 | 3.2 | 1.6 | 1.0 |
| **Ocelot** | 87.6 | 5.2 | 2.4 | 2.2 | 1.8 | 87.1 | 4.3 | — | 1.7 | 2.8 | 3.0 | 79.8 | 8.7 | 4.2 | 3.5 | 1.3 | 1.1 |
| **Southern tiger cat** | 78.8 | 8.3 | 4.8 | 4.7 | 2.6 | 81.8 | 7.4 | — | 2.3 | 4.0 | 3.6 | 70.7 | 10.8 | 8.3 | 5.5 | 1.9 | 1.7 |
| **Bush dog** | 78.1 | 5.9 | 6.8 | 5.1 | 3.8 | 78.6 | 5.6 | — | 3.6 | 6.6 | 4.6 | 75.1 | 7.3 | 7.7 | 6.1 | — | 2.2 |

Those values <1% are not reported. Each species has three ENMs reported: Model 1, Model 2, and the model generated by DeMatteo [1].

**Table 4. The proportion (%) of suitable habitat across the range of potential species richness (0–5 species) within the ~1.5M ha of the northern-central (N-C) zone in Misiones, Argentina, as well as the proportion (%) of total suitable habitat (0–5 species) captured by ~400,000 ha multispecies biological corridor modelled by DeMatteo [1].**

|  | Model 1 | | Model 2 | | DeMatteo [1] | |
|---|---|---|---|---|---|---|
|  | **N-C zone** | **Corridor** | **N-C zone** | **Corridor** | **N-C zone** | **Corridor** |
| **0 species** | 4.5 | — | 12.8 | — | 7.8 | — |
| **1 species** | 14.3 | 1.8 | 14.5 | 1.4 | 13.9 | 2.3 |
| **2 species** | 8.6 | 1.3 | 7.4 | 1.6 | 9.8 | 2.4 |
| **3 species** | 8.9 | 2.4 | 8.0 | 2.3 | 9.1 | 2.9 |
| **4 species** | 10.2 | 3.0 | 16.3 | 8 | 15.3 | 6.1 |
| **5 species** | 53.5 | 18.8 | 41.0 | 15.9 | 44.1 | 14.3 |

Each species has three ENMs compared: Model 1, Model 2, and the model generated by DeMatteo [1].

ENMs capture the overlap and unique characteristics of the five carnivores, as indicated by the finding that the proportion of area occupied by the highest PSR levels (≥4 species), with Model 1 (63.7%) having the highest proportion and Model 2 (57.3%) similar to DeMatteo [1] (59.4%; Table 4).

## Evaluation of DeMatteo [1] corridor

The proportion of suitable habitat captured by the DeMatteo [1] corridor can allow insight into the risk still facing each carnivore, as these areas represent regions lacking formal protection, with all areas intersecting protected areas excluded from these analyses. (Table 2). Model 2 was slightly higher than Model 1 in the proportion of suitable habitat (1.2–5.7%) captured by the corridor, with puma, ocelot, southern tiger cat, and bush dog (Table 2). The exception to this pattern was the jaguar that had a large increase (22.3%) in the proportion of suitable habitat captured by the corridor in Model 2 (Table 2). This trend holds both models compared to the original model, with Model 2 (8.4–41.4%) having a larger proportion of suitable habitat captured by the corridor across all five species than Model 1 (3.8–19.1%; Table 2). Both Model 1 (30.8%) and Model 2 (38.1%) captured a higher average proportion of suitable habitat compared to DeMatteo [1] (20.6%), with Model 2 capturing almost 2x the average proportion of the original model.

The proportion of marginal and optimal habitat captured by the DeMatteo [1] corridor was similar for the bush dog (Table 2). The ocelot and southern tiger cat showed a higher proportion of optimal habitat in the corridor, while the jaguar and puma were the opposite with a higher proportion of marginal habitat captured by the corridor (Table 2). Even with these differences, the spread or difference between the proportion of optimal and marginal habitat captured by the corridor were relatively low in Model 1 (1.9–8.5%) and Model 2 (2.9–7.5%), with the exception of the jaguar (+29.3%) and southern tiger cat (+13.0%), where more marginal habitat was captured in Model 2 (Table 2).

The proportion of the highest PSR levels (≥4 species) captured by the DeMatteo [1] corridor was similar when compared to each other (21.8% and 23.9%) or the original model (20.4%) (Table 4). However, when only the top level of PSR (5 species) captured by the corridor was examined, both Model 1 (18.8%) and Model 2 (15.9%) had higher proportions of suitable habitat than DeMatteo [1] (14.3%)

As with DeMatteo [1], both models had higher levels of native forest (79.1% and 75.6%) versus modified habitats occupying the corridor (Table 5), both models having proportions higher than the original (61.9%). The high proportion of native forest occupying the corridor

**Table 5. Proportion (%) of native forest and modified environments for the northern-central (NC) zone in Misiones, Argentina, and the ~400,000 ha multispecies biological corridor modelled by DeMatteo [1].**

| Variable | N-C zone | Corridor |
|---|---|---|
| **Model 1:** | | |
| Native forest | 72.3 | 79.1 |
| Plantations of pine & eucalyptus | 8.9 | 5.6 |
| Wetlands | — | — |
| Pastures | — | — |
| Herbaceous crops of tobacco & maize | 3.9 | 3.7 |
| Shrub crops of tea & yerba mate | 7.0 | 5.4 |
| Mixed crops with small patches of shrubs & herbaceous plants | 7.0 | 6.1 |
| Naturally bare ground | — | — |
| Water bodies (natural & artificial) | — | — |
| Urban and infrastructure | — | — |
| **Model 2:** | | |
| Native forest | 71 | 75.6 |
| Forest plantations | 10.5 | 8.0 |
| Wetlands | — | — |
| Pastures | — | — |
| Annual agricultural crops | 3.4 | 3.5 |
| Mix of agriculture and pastures | 7.2 | 6.6 |
| No vegetation | 1.0 | — |
| Water | — | — |
| Perennial agricultural crops | 6.1 | 5.6 |
| **DeMatteo [1]** | | |
| Forest | 61.9 | 61.9 |
| Tree plantations | 13.0 | 14.1 |
| Agriculture | 10.7 | 10.1 |
| Mixed use | 6.1 | 7.0 |
| Pasture | 3.0 | 2.4 |
| Bare ground | 1.9 | 2.3 |
| Water | 0.5 | — |
| Urban | 1.0 | 0.4 |
| Campos/grasslands | — | — |
| Unclassified | 1.9 | 1.8 |

Those values <1% are not reported. Each species has three ENMs compared: Model 1, Model 2, and the model generated by DeMatteo [1].

is similar to the proportions of native forest across the five ENMs in the northern-central zone for both models (84.0% and 83.9%), with values again being higher than the original model (75.4%; Table 5). While land use defined categories varied according to the data used in the models, modified land use types grouped into three main categories: forest plantations, agriculture at varying scales, and pastures.

## Identification of regions in need of management strategies

To evaluate the areas of the corridor that need habitat restoration and/or habitat protection, the levels of PSR, as defined by marginal and optimal habitat, were evaluated (Fig 3). This refined analysis goes beyond DeMatteo [1], with the ability to evaluate the quality of habitat

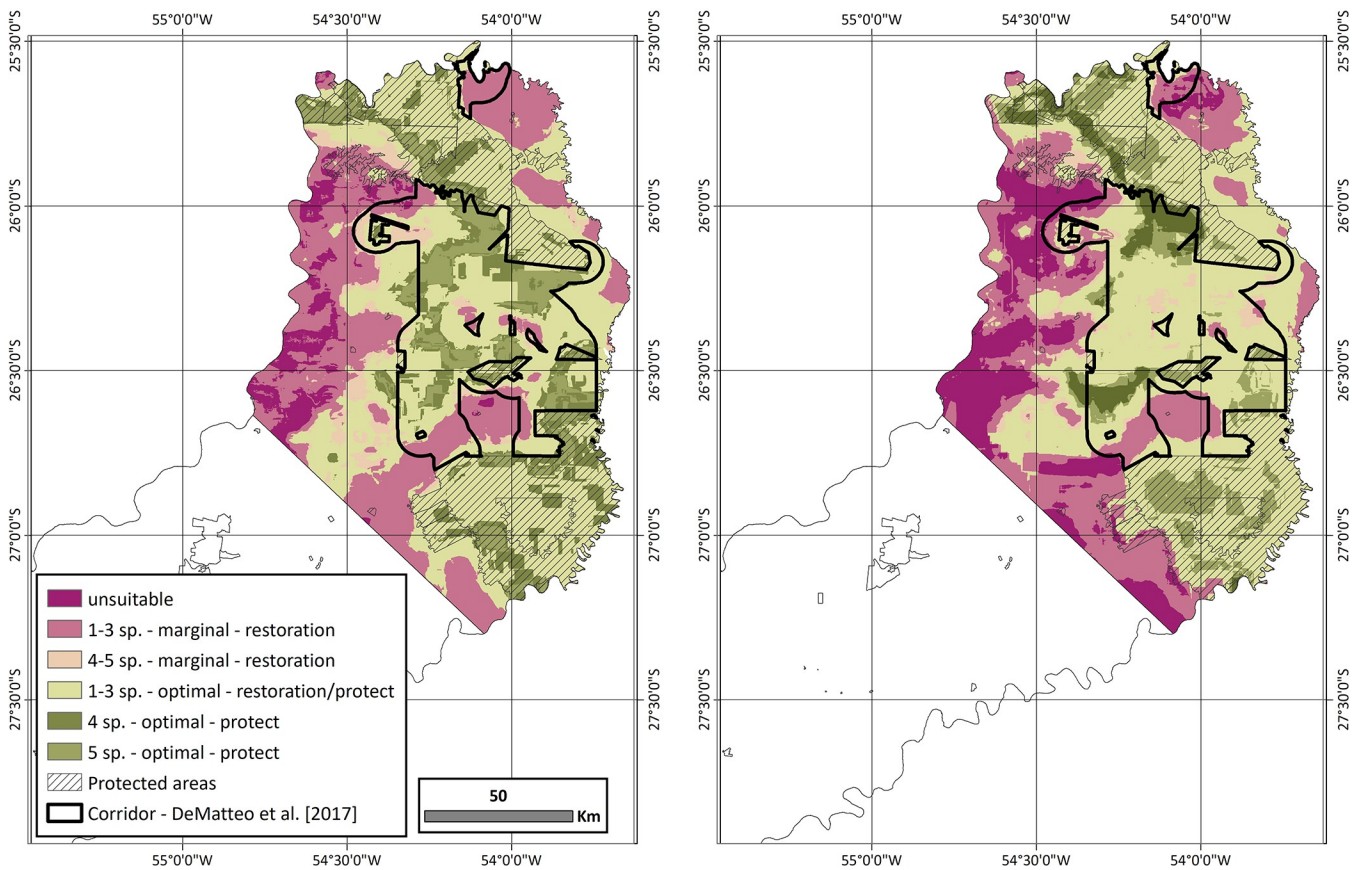

**Fig 3. The potential species richness (PSR) represented as a summary of the proportion (%) of marginal and optimal habitat defined as suitable in the northern-central zone for Model 1 and Model 2.** For each model, the overlap of marginal and optimal habitats was grouped into six classes: 0 species, 1–3 species with marginal habitat, 4–5 species with marginal habitat, 1–3 species with optimal habitat, 4 species with optimal habitat, and 5 species with optimal habitat. Each level is then categorized into the type of land management that is needed to optimized long-term survival of species in the corridor: Habitat restoration, habitat protection, or a combination of the two practices.

for these overlapping species. Across the northern-central zone, both models capture similar totals of suitable habitat (27.7% and 29.3%) and have similar levels of marginal habitat (31.6% and 26.7%), optimal habitat with 1–3 species (44.2% and 45.7%), and optimal habitat with 4–5 species (19.7% and 14.8%) (Table 6). In addition, both models have similar levels of marginal habitat captured in the DeMatteo [1] corridor (19.7% and 17.6%) (Table 6); however, there are slight differences in the proportions of optimal habitat that are contained within the corridor of DeMatteo [1]. The levels of Model 1 are more similar in distribution of optimal habitat with 1–3 species (46.0%) and 4–5 species (34.0%), while Model 2 which has a higher proportion of optimal habitat with 1–3 species (62.3%) compared to 4–5 species (18.2%) (Table 6).

Within the DeMatteo [1], corridor connectors were defined as those regions with the lowest level of PSR and habitat integrity. These regions occupy ~30% of the corridor and were identified as areas that put connectivity between the northern-central zone at risk. This reanalysis with suitable habitat subdivided into marginal and optimal habitat highlights that not all areas within the connector are equal in their needs (Fig 4 and Table 6). While both models require restoration of the marginal habitat that makes up about half of the connectors (58.4% and 51.0%), both models also have a large portion that need a combined management approach. While Model 2 has the majority requiring restoration plus protection (48.4%), Model 1 has

**Table 6. The potential species richness (PSR) with habitat suitability subdivided into marginal and optimal habitat.**

| # Species | Habitat type | Model 1 | | Model 2 | |
|---|---|---|---|---|---|
| | | N-C zone | Corridor | N-C zone | Corridor |
| 0 | unsuitable | 4.5 | 0.3 | 12.8 | 1.9 |
| 1–3 | marginal | 26.8 | 14.6 | 23.3 | 12.0 |
| 4–5 | marginal | 4.8 | 5.1 | 3.4 | 5.6 |
| 1–3 | optimal | 44.2 | 46.0 | 45.7 | 62.3 |
| 4 | optimal | 11.2 | 15.9 | 10.5 | 9.6 |
| 5 | optimal | 8.5 | 18.1 | 4.3 | 8.6 |

For each model two summaries are reported: 1) the proportion (%) of suitable habitat across the range of PSR within the ~1.5M ha of the northern-central (N-C) zone in Misiones, Argentina and 2) the proportion (%) of PSR found within ~400,000 ha multispecies biological corridor modelled by DeMatteo [1]. While marginal habitat is exclusive, within optimal habitat, there are varying levels of marginal habitat that overlap but simplify the categories, only the number of species for optimal habitat are reported.

35.4% in this category and 6.2% in the category of needing protection with optimal habitat and high PSR.

## Discussion

This study used a downstream, multispecies approach where five carnivores, with similar but disparate ecological sensitivities to disturbance, were evaluated individually and then over-lapped to identify common priority areas [38]. As with DeMatteo [1], this expansion beyond a single species approach captured the varied ecological requirements of coexisting carnivores versus the limited requirements of the most restricted species (i.e., jaguar; Fig 1 and Table 2). While the five carnivores vary in their body size, there is a strong intersection in their ecological requirements, with a 41–53.5% (Table 4) overlap in suitable habitat among the five in the northern-central zone. The fact that this overlap is at most ~50% makes it clear that there is existing variation among the five carnivores in the degree of habitat flexibility or sensitivity to human disturbance [20,51,52,87,88]. By developing species-specific ENMs that incorporated frequency of intact or altered habitat (Table 1) in a corresponding area of use, it was possible to quantify the area defined as suitable for each species (Table 2) and differentiate the factors that constrained this suitability (Table 3).

While the two models differed in the proportions of suitable habitat and associated vegetation, when compared to each other and to DeMatteo [1], the general patterns were consistent across the five carnivores. In contrast to the highly-restricted jaguar, the southern tiger cat and the bush dog had the largest suitable areas in the northern-central zone of Misiones, followed by the ocelot and puma (Fig 2). A similar pattern is seen among the five carnivores in the proportion of suitable habitat outside of protected areas, which are primarily composed of native forest, with the southern tiger cat and bush dog almost 2x that of the jaguar (Table 2). The jaguar's low proportion of suitable habitat outside of protected areas (21.8–30.2%) provides insight into its preference to proximity to large tracts of forest versus altered habitat. Similarly, the puma and ocelot are associated with higher levels of forest, which explains the slightly lower levels of suitable habitat for these species in the northern-central zone and outside of protected areas compared to the levels found in the southern tiger cat and bush dog.

When these levels of suitability are subdivided, it is the two species at the extremes in the proportion of suitable habitat, jaguar and bush dog, that have the highest proportions of

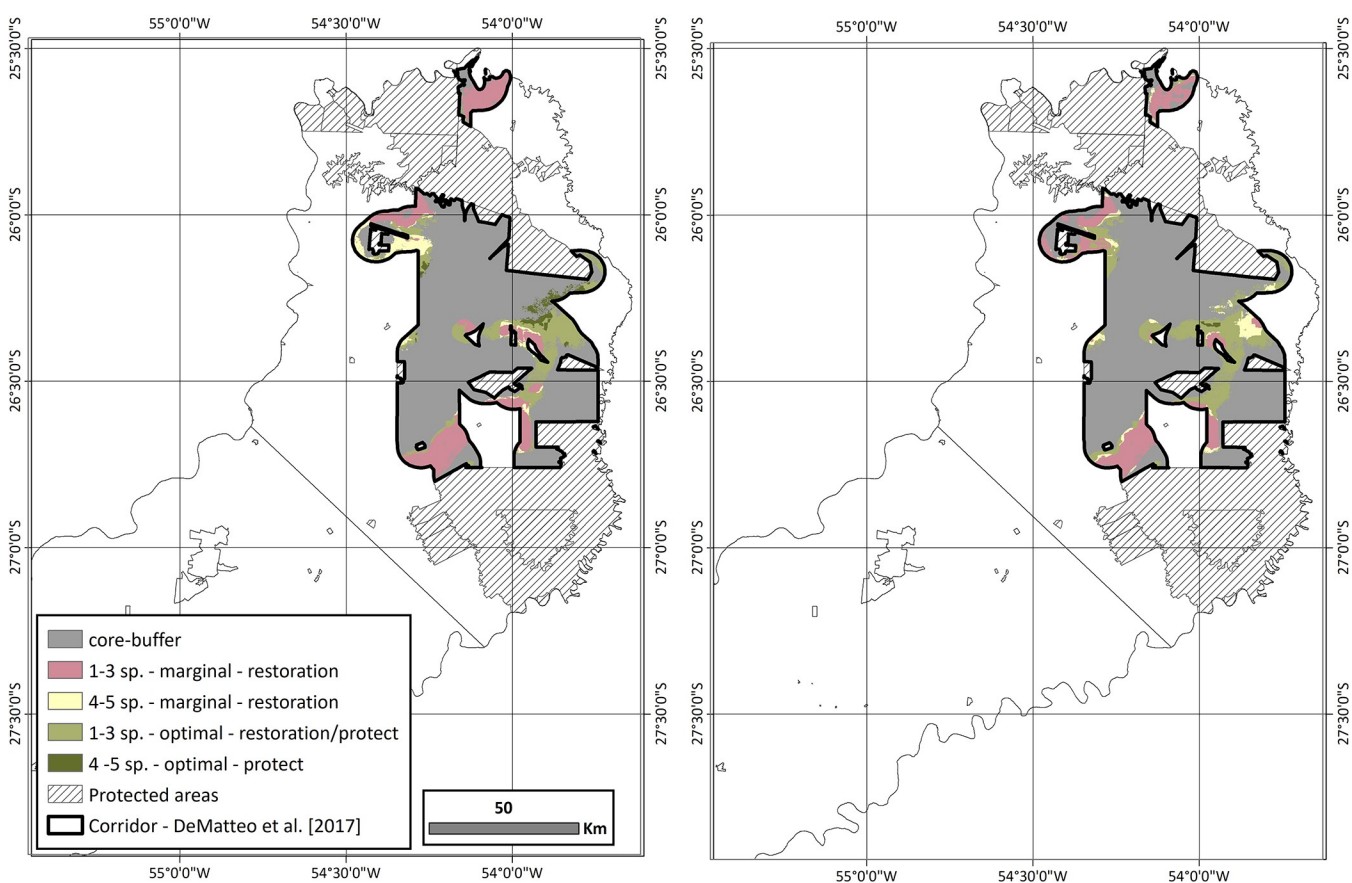

**Fig 4. The potential species richness (PSR) is represented as a summary of the proportion (%) of marginal and optimal habitat defined as suitable in the northern-central zone for Model 1 and Model 2.** Presented is the overlap of the potential species richness (PSR) with defined marginal and optimal habitats for each model with the DeMatteo [1] corridor, specifically the connectors or those areas defined as having the lowest levels of PSR and habitat integrity. Areas in the connectors are defined by the type of habitat (marginal or optimal), the number of overlapping species (1–5), and whether habitat restoration and/or protection are needed. The remainder of the corridor is generalized as the core buffer, as it corresponds to the DeMatteo [1] model.

marginal habitat in the northern-central zone. This is compared to the moderate ocelot that has higher proportions of optimal habitat. When narrowed to the areas outside of protected areas, the jaguar and bush dog are joined by the puma in sharing the highest levels of marginal habitat. Together these findings suggest that the long-term survival of the jaguar in the northern-central zone is at risk with areas outside of protected areas of only marginal habitat quality. While the bush dog appears to be more flexible, it is critical to remember that while it is a species that exists across varied habitats, these habitats typically provide lower quantities of prey and a higher risk for exposure to domestic dog diseases [81,89]. The same is true for the puma, which is a high risk of poaching when crossing open areas near human-occupied areas. Interestingly, the ocelot, which is associated with higher levels of forest in this study and others [88,90], still shows a degree of flexibility outside of protected areas with high levels of optimal habitat, which suggests it might not be as restricted in the variety of habitats it can occupy [91].

Using species-specific ENMs, DeMatteo [1] developed resistance layers for each carnivore, which when merged provided a composite layer of potential movement across the landscape [38]. Using this combined resistance map for the five species, DeMatteo [1] centered the ~400,000 ha corridor on areas with the lowest ecological cost for the five carnivores. However, unlike DeMatteo [1] where the jaguar was the most restricted, as reflected by the lowest

proportion (13.9%) of suitable habitat in the corridor compared to the other four carnivores (21.0–24.3%), the current models show similar proportions of suitable habitat contained within the corridor for each of the five carnivores (Table 2). Both models capture a higher average proportion of suitable habitat (1.5-2x) across the carnivores compared to DeMatteo [1]. While the habitat flexibility of the bush dog and puma are reflected in the higher proportions of marginal habitat contained in the corridor, the risk to the jaguar's mobility between protected areas is echoed with its higher proportion of marginal habitat. The ocelot's comparatively higher level of optimal habitat in the corridor, again puts into question, its potential habitat flexibility [91].

Given that the DeMatteo [1] corridor connects existing suitable habitats in a heterogeneous landscape versus man-made structures as connectors, an investment to conserve this ~400,000 ha area would be considered worthwhile [92]. However, the process to determine key conservation targets must balance multiple factors, including species richness and cost allocation [93], which inevitably overlaps with finding a balance between the contrasting needs of wildlife and humans [94–98], especially as humans continue to expand into the backyards of wildlife [99–102]. While DeMatteo [1] used overlapping habitat suitability and degree of habitat fragmentation to differentiate areas that need varying degrees of protection and habitat restoration, these analyses expanded beyond general suitability and into varying levels of suitability. While both models were similar in the proportion of area needing restoration (high PSR with marginal habitat), a combined approach of restoration-protection (low PSR with optimal habitat), or protection (high PSR with optimal habitat), this refined analysis highlighted some key differences (Table 6). Specifically, when focused on the areas defined as connectors or those areas with the lowest level of PSR and habitat integrity [1]. While these refined analyses found that ~50% of the connectors were composed of marginal habitat, the remainder corresponds to optimal habitat with varying levels of PSR (Fig 4 and Table 6). This means that management in these connectors must include work with the existing landowners [103–105] and extend beyond restoration and include protection of the optimal habitat and the species that occupy it. These results suggest there remains a high degree of connection between the northern-central zones of Misiones but without immediate action to protect the optimal habitat in the eastern portion of the connector, the varying levels of PSR are put at risk.

One question that remains is which of the two models more accurately reflects the situation in northern-central Misiones and the corridor of DeMatteo [1]. While one might argue that Model 1 (2015) is based on older data where higher levels of native forest would exist. An initial comparison of the habitat associated with individual scats between the two years supports this idea with all five species having lower levels of forest in Model 2 compared to Model 1, with similar levels in the four felids (84.7–89.1), but lower levels in the bush dog. However, further analyses that examine other types of altered habitat suggests that the difference in years does not explain all the differences. Instead, there appears to be a discrepancy in how the pixels are defined versus an actual shift in habitat quality (e.g., intact versus altered). For example, with the southern tiger cat, Model 2 aligns with only 64.1% forestry plantations, with the remaining proportion (35.9%) including areas with no vegetation, forest, and annual agriculture crops. These differences expand with other habitat types, with Model 2 aligning with only 16.2% shrub crops, 6.7% herbaceous crops, and 5% mixed use of Model 1, with the remaining portion including a mix of native forest and other types of altered habitats. The terrain in Misiones outside of protected areas is extremely heterogeneous with a mix of habitat types in close proximity to each other. What was historically shrub crops of only yerba, are now frequently a mix of yerba planted among a network of native trees. While some pine plantations are composed of densely planted trees, others are more widely spaced allowing a native forest to become established. In some cases, additional ground truthing for defined areas of different

vegetation types may help but in others the overlap between pixel values may be blurred making a clean definition of one type or the other impossible or varied between interpretations. Despite the differences in the individual layers used in each model, the final results were comparable, suggesting the definition of intact or altered is consistent even if the defined type of habitat varies.

While this model was comprehensive with a multispecies approach that incorporated anthropogenic variation across the landscape [38], future models could expand in several directions. First, the Green Corridor of Misiones expands into the central-southern zone of Misiones. While climatic conditions are similar in this region, the degree of ongoing habitat conversion, poaching, and growing populations are magnitudes higher. Understanding how species distributions shift in their unique and overlapping values is essential to developing conservation strategies in this region. This type of analysis can also extend outside of Misiones, as the species in this study have ranges across the Neotropics. Therefore, the results from this study can be expanded to understand the connectivity between countries, with analyses at the species or community level [106]. Second, analyses could expand to include other variables, such as the overlap of prey species and presence of illegal hunting. Both of these factors directly and indirectly affect the distribution of carnivores. Finally, these types of analyses could explore how future climate predictions could shift the distribution of these five carnivores [107–110]. While there northern-central zones of Misiones show little variation in climate, it is evident that what was once considered 'normal' is shifting. The range of seasonality, in both temperatures and rainfall, is more variable, leading to shifts in water availability, soil moisture levels, and risk of fire. Given the distribution of these five carnivores beyond the boundaries of Misiones and the ever-expanding human footprint [2,3], there is a need to understand how ongoing habitat fragmentation and shifts in climate can affect the long-term survival of these species.

## Supporting information

**S1 Appendix. Details of the 1085 scat swabs with confirmed species identity.** For the 1034 felid samples and 51 bush dog samples, the location and zone (North-Central) are summed by species. For protected areas, the total area is reported in parentheses.
(DOCX)

## Acknowledgments

The Ministerio de Ecología y Recursos Naturales Renovables of Misiones (MEyRNR) and the Administración de Parques Nacionales of Argentina provided permits and help with field planning. The MEyRNR provided housing. Multiple US zoos (Birmingham Zoo, Cincinnati Zoo, Detroit Zoo, Exotic Feline Breeding Compound, Little Rock Zoo, Palm Beach Zoo, Saint Louis Zoo, San Antonio Zoo, and Sedgwick County Zoo) and three field researchers (C. Vynne, C. Wultsch, and N. Songsasen) provided detection dog training samples. Sincere thanks to the numerous Argentinean students who assisted in the field, the provincial/national park guards, NGO's, local conservationists, private land/reserve owners, and colleagues that helped with various aspects of this project in the field and in the lab. Thank you to PackLeader Conservation Detection Dogs for guidance and to all members of the Olsen/Eggert Labs for their support. Of course, a special thanks to "T" whose amazing work ethic made it all possible.

## Author Contributions

**Conceptualization:** Karen E. DeMatteo, Miguel A. Rinas, Carina F. Argüelles.

**Formal analysis:** Karen E. DeMatteo.

**Funding acquisition:** Karen E. DeMatteo, Miguel A. Rinas, Carina F. Argüelles.

**Investigation:** Karen E. DeMatteo, Orlando M. Escalante, Daiana M. Ibañez Alegre, Miguel A. Rinas, Delfina Sotorres, Carina F. Argüelles.

**Methodology:** Karen E. DeMatteo.

**Project administration:** Karen E. DeMatteo, Miguel A. Rinas, Carina F. Argüelles.

**Resources:** Karen E. DeMatteo, Miguel A. Rinas, Carina F. Argüelles.

**Supervision:** Karen E. DeMatteo, Miguel A. Rinas, Carina F. Argüelles.

**Validation:** Karen E. DeMatteo.

**Visualization:** Karen E. DeMatteo, Miguel A. Rinas.

**Writing – original draft:** Karen E. DeMatteo.

**Writing – review & editing:** Karen E. DeMatteo, Orlando M. Escalante, Daiana M. Ibañez Alegre, Miguel A. Rinas, Delfina Sotorres, Carina F. Argüelles.

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
