## [Decision Letter · Decision Letter 0]

2 Nov 2022

PONE-D-22-23711

A multispecies corridor in a fragmented landscape: evaluating effectiveness and identifying high-priority target areas

PLOS ONE

Dear Dr. DeMatteo,

Thank you for submitting your manuscript to PLOS ONE. After careful consideration, we feel that it has merit but does not fully meet PLOS ONE’s publication criteria as it currently stands. Therefore, we invite you to submit a revised version of the manuscript that addresses the points raised during the review process.

Specifically, the reviewer has provided some minor comments which need to be addressed. 

We look forward to receiving your revised manuscript.

Kind regards,

Daniel de Paiva Silva, Ph.D.

Academic Editor

PLOS ONE

Journal Requirements:

1. We noticed you have some minor occurrence of overlapping text with the following previous publication(s), which needs to be addressed:

https://journals.plos.org/plosone/article?id=10.1371%2Fjournal.pone.0183648

In your revision ensure you cite all your sources (including your own works), and quote or rephrase any duplicated text outside the methods section. Further consideration is dependent on these concerns being addressed.

3. We note that Figures 1a, 1b, 2a, 2b, 3a, 3b, 4a and 4b in your submission contain map images which may be copyrighted. All PLOS content is published under the Creative Commons Attribution License (CC BY 4.0), which means that the manuscript, images, and Supporting Information files will be freely available online, and any third party is permitted to access, download, copy, distribute, and use these materials in any way, even commercially, with proper attribution. For these reasons, we cannot publish previously copyrighted maps or satellite images created using proprietary data, such as Google software (Google Maps, Street View, and Earth). For more information, see our copyright guidelines: http://journals.plos.org/plosone/s/licenses-and-copyright.

a. You may seek permission from the original copyright holder of Figures 1a, 1b, 2a, 2b, 3a, 3b, 4a and 4b to publish the content specifically under the CC BY 4.0 license.  

Additional Editor Comments (if provided):

Reviewers' comments:

Reviewer's Responses to Questions

**Comments to the Author**

1. Is the manuscript technically sound, and do the data support the conclusions?

Reviewer #1: Yes

2. Has the statistical analysis been performed appropriately and rigorously? 

Reviewer #1: Yes

3. Have the authors made all data underlying the findings in their manuscript fully available?

Reviewer #1: Yes

4. Is the manuscript presented in an intelligible fashion and written in standard English?

Reviewer #1: Yes

5. Review Comments to the Author

Reviewer #1: Dear Authors,

Congratulations on producing a great piece of work. The extensive surveys are to be commended and the dataset generated is incredible. I found the manuscript to be an enjoyable read, and the methods and anlayses appropriate. Well done! I recommend your work should be accepted with only minor revisions (see attached).

Kind regards,

Ana Gracanin

PhD Candidate

Centre for Sustainable Ecosystem Solutions

School of Earth, Atmospheric and Life Sciences

Faculty of Science, Medicine & Health

University of Wollongong NSW 2522 Australia

6. PLOS authors have the option to publish the peer review history of their article (what does this mean?). If published, this will include your full peer review and any attached files.

Reviewer #1: No

---

## [Author Response · Author response to Decision Letter 0]

8 Dec 2022

Reviewer comments:

All suggested word changes in the document were made, with a few exceptions. A general note: in the references at the end of the manuscript and in the appendix, there were multiple notes for formatting in Italian; however, no actual changes to the text were noted, so no changes were made in this section. The exceptions were:

1) Line 85: The use of “its” is appropriate in this case, as it indicated possessive, with Misiones being the region’s National Capital of Biodiversity. “It’s” would be incorrect, as an abbreviation of “it is” is not appropriate here. 

2) Line 219: The use of “was” versus “were” is correct in this instance, as it is referring to the ‘neighborhood scale’, which is singular. 

3) Line 240: The use of “was” versus “were” is correct in this instance, as it is referring to the ‘efficacy’, which is singular. 

4) Line 267: The use of “fit” versus “fits” is appropriate and follows the rest of the paragraph where “fit” is considered appropriate. 

5) Line 303: The text was not changed to “suitability levels reported”, as this would remove consistency for future sections in the Discussion (page 28 and page 30) where the results are discussed relative to “levels of suitability”. The change is a wording preference versus a change of meaning. 

Editor comments:

1) Overlapping text: Per an email on 29 November 2022, Publication Assistant Alexis Miller noted that a review of the text found the overlap of text to be within the methods section. Therefore, the text of manuscript is acceptable under PLOS policy (Case 07798633). 

2) An ethics statement was added, noting no verbal or written permission is required by the Saint Louis Zoo’s IACUC committee. 

3) Question about copyrighted map images: Per an email on 7 December 2022, Publication Assistant Alexis Miller noted that a review of the journal policy did not find any potential copyright concerns regarding the figures in question. Therefore, the figures in the manuscript are acceptable under PLOS policy (Case 07798633).

4) Review of reference list: A review was made and no citations have been retracted.

---

## [Decision Letter · Decision Letter 1]

6 Mar 2023

A multispecies corridor in a fragmented landscape: evaluating effectiveness and identifying high-priority target areas

PONE-D-22-23711R1

Dear Dr. DeMatteo,

We’re pleased to inform you that your manuscript has been judged scientifically suitable for publication and will be formally accepted for publication once it meets all outstanding technical requirements.

Kind regards,

Daniel de Paiva Silva, Ph.D.

Academic Editor

PLOS ONE

Additional Editor Comments (optional):

Dear Dr. DeMatteo,

I am pleasured to inform you and your co-authors that your manuscript has been formally accepted for publication in PLoS One. Both reviewers agree that improvements were made to the new version of the manuscript and one of them suggested that new figures with better resolution are prepared and added to the manuscript. Once again, congratulations on your efforts to improve the text.

Sincerely,

Daniel Silva

Reviewers' comments:

Reviewer's Responses to Questions

**Comments to the Author**

1. If the authors have adequately addressed your comments raised in a previous round of review and you feel that this manuscript is now acceptable for publication, you may indicate that here to bypass the “Comments to the Author” section, enter your conflict of interest statement in the “Confidential to Editor” section, and submit your "Accept" recommendation.

Reviewer #1: All comments have been addressed

Reviewer #2: All comments have been addressed

2. Is the manuscript technically sound, and do the data support the conclusions?

Reviewer #1: Yes

Reviewer #2: Yes

3. Has the statistical analysis been performed appropriately and rigorously? 

Reviewer #1: Yes

Reviewer #2: Yes

4. Have the authors made all data underlying the findings in their manuscript fully available?

Reviewer #1: Yes

Reviewer #2: Yes

5. Is the manuscript presented in an intelligible fashion and written in standard English?

Reviewer #1: Yes

Reviewer #2: Yes

6. Review Comments to the Author

Reviewer #1: (No Response)

Reviewer #2: The manuscript addresses an important issue and emphasizes the importance of using multispecies corridor to select target areas in fragmented landscapes that require different management strategies of restoration, preservation, or both.

I appreciate the opportunity of reading your manuscript. Methods analysis were carried out carefully, reflecting the expertise of the authors. Please, I just suggest to check the resolution of the figures to have the best quality to final publication.

Finally, I congratulate you on the presentation of results and discussion, and the study of such relevant topic of finding solutions to prioritize conservation areas considering several species. In my opinion, the manuscript is ready to be published.

7. PLOS authors have the option to publish the peer review history of their article (what does this mean?). If published, this will include your full peer review and any attached files.

Reviewer #1: No

Reviewer #2: No

---

## [Editor Report · Acceptance letter]

5 Apr 2023

PONE-D-22-23711R1 

A multispecies corridor in a fragmented landscape: evaluating effectiveness and identifying high-priority target areas 

Dear Dr. DeMatteo:

I'm pleased to inform you that your manuscript has been deemed suitable for publication in PLOS ONE. Congratulations! Your manuscript is now with our production department. 

Kind regards, 

on behalf of

Dr. Daniel de Paiva Silva 

Academic Editor

PLOS ONE